# Elucidating the Subcellular Localization of GLRaV-3 Proteins Encoded by the Unique Gene Block in *N. benthamiana* Suggests Implications on Plant Host Suppression

**DOI:** 10.3390/biom14080977

**Published:** 2024-08-09

**Authors:** Patrick Lameront, Mehdi Shabanian, Laura M. J. Currie, Catherine Fust, Caihong Li, Alyssa Clews, Baozhong Meng

**Affiliations:** Department of Molecular and Cellular Biology, University of Guelph, Guelph, ON N1G 2W1, Canada; shabania@uoguelph.ca (M.S.); lcurri03@uoguelph.ca (L.M.J.C.); cfust@uoguelph.ca (C.F.); caihong@uoguelph.ca (C.L.); clewsa@uoguelph.ca (A.C.); bmeng@uoguelph.ca (B.M.)

**Keywords:** p21, p20A, p20B, viral suppressor, RNA silencing, grapevine, leafroll disease

## Abstract

Grapevine leafroll-associated virus 3 (GLRaV-3) is a formidable threat to the stability of the global grape and wine industries. It is the primary etiological agent of grapevine leafroll disease (GLD) and significantly impairs vine health, fruit quality, and yield. GLRaV-3 is a member of the genus *Ampelovirus*, *Closteroviridae* family. Viral genes within the 3′ proximal unique gene blocks (UGB) remain highly variable and poorly understood. The UGBs of *Closteroviridae* viruses include diverse open reading frames (ORFs) that have been shown to contribute to viral functions such as the suppression of the host RNA silencing defense response and systemic viral spread. This study investigates the role of GLRaV-3 ORF8, ORF9, and ORF10, which encode the proteins p21, p20A, and p20B, respectively. These genes represent largely unexplored facets of the GLRaV-3 genome. Here, we visualize the subcellular localization of wildtype and mutagenized GLRaV-3 ORFs 8, 9, and 10, transiently expressed in *Nicotiana benthamiana*. Our results indicate that p21 localizes to the cytosol, p20A associates with microtubules, and p20B is trafficked into the nucleus to carry out the suppression of host RNA silencing. The findings presented herein provide a foundation for future research aimed at the characterization of the functions of these ORFs. In the long run, it would also facilitate the development of innovative strategies to understand GLRaV-3, mitigate its spread, and impacts on grapevines and the global wine industry.

## 1. Introduction

The grapevine leafroll-associated virus 3 (GLRaV-3), the chief agent associated with grapevine leafroll disease (GLRD), is a source of significant concern for viticulture and wine industry worldwide. This virus and the resulting disease pose a major threat to grapevine vitality, yield, and quality [1,2]. GLRD infection induces several phenotypic traits including altered fruit development and quality as well as leaf morphology. Infected grapevine leaves exhibit a downward rolling of leaf margins and overall discoloration, specifically, the transition to red to dark purple interveinal regions on dark-berried cultivars and leaf chlorosis on white-berried cultivates [3]. The virus is transmitted through infected propagation materials and *Pseudococcidae* mealybugs [4,5]. In the progressive stages of GLRaV-3 infection, berries show an overall decline in development, ultimately affecting the gross amount and quality of wines and juices [6]. While the severity of GLD symptoms is variable and dependent on the cultivar, viral titer, and environmental conditions, infected vines have recorded yield losses of up to 50% [2]. This substantial reduction in grapevine productivity can compound over growing cycles, affecting the long-term sustainability of a vineyard, especially if detection and management strategies are not put in place [7].

GLRaV-3 infection, replication, and trafficking are limited to phloem tissue and impair source-to-sink photosynthate transport from leaves to berries [8]. Disruption in sugar flow results in elevated sugar concentration in leaves and increased allocation of resources to flavonoid biosynthesis pathways [9]. This induces the characteristic color change in the leaves of GLRaV-3-infected dark-berried grapevine cultivars as anthocyanidins are converted to anthocyanins in the presence of excess sugar. Phloem tissue of grapevines infected with GLRaV-3 undergoes significant cytopathological alterations, including the presence of vesiculated mitochondria [10]. Conversely, the berries receive a diminished sugar supply, leading to a reduction in pigmentation, size, amino acid, and metabolite composition. The amino acid composition affects the rate and completion of fermentation [11,12]. GLRaV-3 is a member of the genus *Ampelovirus*, of the *Closteroviridae* family of phloem-limited plant viruses, containing some of the largest positive-sense single-stranded RNA (+ssRNA) genomes [13]. Characterized by bipolar flexible helically-constructed filaments, these viruses have virions 12 nm in diameter and range between 650 nm and 2200 nm in length [14]. The intricate expression strategies of members of the *Closteroviridae* involve proteolytic processing, ribosomal frameshifting, and the production of sub-genomic mRNAs to enhance viral coding capacity and regulate genome expression [3]. Within this family, GLRaV-3 holds a distinctive place, being the prototypical member of the *Makeovers* genus.

The approximately 18.5 kb-long genome of GLRaV-3 harbors 12–13 ORFs that encode various viral proteins (Figure 1A) [7]. While the functions of many of these ORFs have been inferred based on similarities in their sequence and genomic location to other viruses within the *Closteroviridae* family, few have been experimentally confirmed [15]. The replication gene block (RGB)—consisting of ORF1a and ORF1b—plays a pivotal role in viral transcription and genome replication. Utilizing a ribosomal frameshift, these ORFs are translated into two polyproteins that are subsequently processed to form a leader protease, a methyltransferase, a helicase, and an RNA-dependent RNA polymerase (RdRP) [3]. The translation product of ORF1a contains an AlkB domain that is hypothesized to be involved in RNA repair through demethylation [16,17]. Following ORF1a and ORF1b, ORF2 theoretically encodes a small protein but this genomic feature is only present in select isolates of the virus and likely does not play a necessary role in the viral infection cycle [18]. 

A hallmark of *Closteroviridae* genomes is the quintuple gene block (QGB) (ORFs 3–7), which is central to the virus life cycle [19]. ORF3 encodes a small transmembrane protein that is targeted to the endoplasmic reticulum (ER) [20]. ORF4 encodes the molecular chaperone protein, Hsp70h, homologous to cellular Hsp70; ORF5 encodes a 60 kDa protein; ORF6 encodes the major coat protein (CP), while ORF7 encodes the minor capsid protein (CPm) [21,22]. Genes in this region cooperate in virion and head assembly. Taken together, the virion head structure is a molecular device that enables the intercellular transport of virions through the plasmodesmata [19].

However, GLRaV-3 strays from the conventional genome structure of *Closteroviridae* with the addition of ORF8, ORF9, and ORF10, which encode proteins named p21, p20A, and p20B, respectively. While other *Closteroviridae* members contain unique genomic regions at their 3′ end, the structure and functions of these gene products remain largely unexplored [23,24,25]. Subgenomic mRNAs for ORF8, ORF9, and ORF10 are among the most abundant in GLRaV-3-infected grapevines, indicating a potentially important role [26]. p20B has also been found to self-interact and evidence suggests an interaction with Hsp70h, the major and minor coat proteins [27]. p20B has also been identified as a viral RNA silencing suppressor (VRSS) in the herbaceous model plant *N. benthamiana* and a potential determinant of pathogenicity [28]. The functions of proteins encoded by ORF8 and ORF9 remain unknown. ORF11 and ORF12 exist on the 3′ terminus of the genome and are significantly diverse in sequence and size among GLRaV-3 isolates; thus, a conserved function is unlikely [29].

RNA silencing, also known as RNA interference (RNAi), is used to combat viral infections and to regulate the expression of genes at critical stages of development in plants and invertebrates [30]. This process was expected to target and degrade viral RNA, thus hindering replication and the spread of the invading virus both intracellularly and intercellularly [31]. dsRNA, the replicative intermediate produced during genome replication of RNA viruses, is recognized by Dicer-like (DCL) proteins in the cytosol. DCL proteins enzymatically process dsRNA into small interfering dsRNA (siRNA), typically 21–24 base pairs in length [32]. siRNAs are loaded into a class of proteins called Argonaute proteins (AGO) [33]. Loaded AGO proteins are cycled into the nucleus where the complex is further processed into an RNA-induced silencing complex (RISC); Nuclear Hsp90 and various co-chaperones eject the passenger strand of ssRNA from the complex while the guide strand is retained. Processing steps within the nucleus are crucial to RISC maturation and RNA silencing [34,35]. Once mature, RISC is returned to the cytosol where the guide strand RNA, contained within each RISC, guides the complex to complementary viral RNA sequences. The RISC specifically recognizes and binds to viral ssRNA, cleaving the viral genetic material. Thus, RNA silencing targets viral RNA for degradation, preventing the synthesis of viral proteins crucial for the viral life cycle and the generation of new virions. Furthermore, the antiviral signal is spread through the movement of siRNAs to neighboring cells and even to distant parts of the plant [36,37].

In the context of plant-virus interactions, host RNA silencing serves as a potent antiviral defense strategy. However, many plant viruses have co-evolved mechanisms to counteract the host RNA silencing response [38]. VRSS proteins are encoded by numerous viruses to inhibit different steps of the RNA silencing pathway. These include interference with RNA silencing by disrupting the sensing and slicing of dsRNA, blocking RISC assembling, sequestering siRNA, targeting AGO proteins, and suppressing the amplification of antiviral silencing signals dispersed beyond the advancing infection front. Ultimately, these strategies allow viruses to evade host defense and establish successful infections [38]. Understanding the interplay between RNA silencing as a plant defense mechanism and viral counterstrategies is crucial for deciphering the molecular arms race between plants and viruses. Two of the well-studied VRSS proteins with unique mechanisms are p19 and 2b, encoded by tombusviruses and cucumber mosaic virus, respectively. p19 forms homodimers capable of sequestering double-stranded siRNA. Each p19 subunit forms an arm of a molecular caliper set that measures, distinguishes, and binds siRNA, preventing its loading into AGO proteins [39,40]. In a separate strategy, 2b interacts with the PAZ domain of AGO1 to inhibit the slicing of siRNA. Furthermore, 2b colocalizes with AGO1 in cytoplasmic foci and the nucleus [41].

The aim of this investigation is to study the subcellular localization and behavior of three of the proteins encoded by the UGB of GLRaV-3, namely ORFs 8–10, as the first step in unraveling the function of these viral proteins. The complex genome of GLRaV-3 encodes unique viral proteins likely involved in viral movement, modulation of the host defense system, and pathogenesis. Understanding the genome of GLRaV-3, the function of its proteins, and their regulation is imperative for the development of effective strategies to control the spread and impact of this virus on grapevines. This study aims to bridge the existing knowledge gap by examining the localization strategies and mechanisms of molecular association between unique gene products of GLRaV-3. By the ectopic expression of each p21, p20A, and p20B tagged with Enhanced Green Fluorescent Protein (EGFP), here, for the first time, we report their localization to the cytosol, microtubules, and nucleus, respectively. These findings narrow down the potential mechanisms through which these viral proteins suppress and manipulate the host cell.

## 2. Materials and Methods

In silico Analysis of p21, p20A, and p20B

The amino acid sequences of p21, p20A, and p20B from GLRaV-3 isolate 623 were uploaded to Colabfold to predict their 3D structures [42]. Predicted protein structures were analyzed by distance matrix alignment (Dali) to compare structural similarity with experimentally determined protein structures deposited to the Protein Data Base (PDB) [43]. Adaptive Poisson-Boltzman Solver (APBS) software (APBS Tools2.1) was used to generate surface electrostatics for each structure [44]. The predicted structures were submitted to Galaxy Homomer to generate homo-oligomer conformations using template-based modeling and ab initio docking [45]. The amino acid sequences were submitted to Plant-mPLoc, a subcellular localization predictor specific for plant proteins [46]. To follow up, the amino acid sequence of p20B was submitted to Identification Nucleus Signal Peptide (INSP) software, http://www.csbio.sjtu.edu.cn/bioinf/INSP/ (accessed on 29 July 2024) [47].

Plant Material and Growth Conditions.

*N. benthamiana* was chosen as the model system for our study as it is amenable to agrobacterium-mediated genetic transformation, is susceptibile to a wide range of pathogens, and grows rapidly in controlled environments [48]. Wildtype *N. benthamiana* plants were grown from seeds in ProMix BX (no mycorrhizae) soil and grown at 24 °C, 150 µmol/m^2^/s PPFD, 37% relative humidity, with a 16-h light/8-h dark photoperiod. Seeds were sewn in a pot of moistened ProMix BX with a humidity cover until germination. Seedlings were then transplanted to individual pots of moistened ProMix BX and placed in a tray with a humidity cover. Once seedlings grew to touch the humidity cover, roughly 2 weeks after potting, the cover was removed and trays of plants were watered every 2 to 3 days afterward. Plants were grown until they reached the 4–6 leaf stage, roughly 6 weeks after seeds were sewn. Plants at this stage were used for agroinfiltration experiments.

Plasmid Constructs and *Agrobacterium tumefaciens* strains

The DNA cloning process commenced with the amplification of GLRaV-3 ORF 8, 9, and 10 individually with gene-specific primers from full-length GLRaV-3 isolate 623 infectious clones using PCR. The PCR reactions were executed using a high-fidelity DNA PaCeR polymerase (GeneBioSystems, Burlington, ON, Canada), used according to the manufacturer’s instructions. Primers were designed to include an NcoI restriction site in the forward primer and a Kpn1 restriction site in the reverse primer (primers shown in Table 1). Subsequently, the amplified fragments of ORF 8, 9, and 10 were purified using PureLink™ PCR Purification Kit (Thermo Fisher Scientific, Branchburg, NJ, USA). Amplified fragments and the expression vectors pRTL3-EGFP and pRTL3-mRFP were subjected to restriction digestion with the FastDigest restriction enzymes (Thermo Scientific, Waltham, MA, USA) according to the manufacturer’s instructions. The digested DNA fragments were ligated into the vectors using T4 DNA ligase (Thermo Scientific) according to the manufacturer’s instructions (Figure 1B). Following successful cloning into the expression vectors, the expression cassettes and the binary vector pCAMBIA-0380 were digested with FastDigest HindIII and BcuI restriction enzymes. The digested DNA fragments were ligated into the vectors using T4 DNA ligase. Further confirmation was achieved through Sanger sequencing at the Advanced Analysis Centre (AAC) at the University of Guelph. Once the constructs were validated, plasmid DNA was transformed into *Agrobacterium tumefaciens* strains EHA105 or GV3101.

Agroinfiltration Procedure

Agroinfiltration was carried out using a needleless syringe method [48]. Briefly, *A. tumefaciens* strains carrying the desired plasmid constructs were cultured in Luria–Bertani (LB) broth at room temperature for 16 h and pelleted and resuspended in an infiltration buffer containing 10 mM MgCl_2_, 10 mM MES (pH 5.6), and 150 μM acetosyringone. The optical density at 600 nm (OD_600_) was adjusted to a desired amount dependent on the *A. tumefaciens* strain (Appendix A). The bacterial suspensions were incubated at room temperature for 16 h prior in an infiltration buffer to allow time for the induction of vir genes. Agroinfiltration experiments were performed by injecting *N. benthamiana* leaves with *A. tumefaciens* suspended in an infiltration buffer. A. tumefaciens transformed with different constructs were infiltrated alone and in combination.

Site-Directed Mutagenesis

Quick-change site-directed mutagenesis [49] was employed to introduce specific mutations in the putative nuclear trafficking signal of p20B-EGFP-GUS between positions 109 to 120 (primers shown in Table 1). The ΔNLS2 primer set was employed to mutate two lysine residues at positions 118 and 119 to alanine (Ala) and the ΔNLSε primer set was employed to mutate the arginine (Arg) and lysine (Lys), residues between positions 112 to 115 to alanine.

Microscopy and Subcellular Localization.

Agrobacterium-infiltrated *N. benthamiana* leaves were processed at 2 days post infiltration (dpi) for CLSM imaging. Micrographs of *N. benthamiana* leaves were acquired at the Advanced Analysis Centre at the University of Guelph using a Leica DM 6000B microscope (Leica, Wetzlar, Germany) connected to a Leica TCS SP5 confocal laser scanning microscopy equipped with a 63× Plan Apochromat oil-immersion objective (numerical aperture [NA]¼1.32), TCS SP2 scanning head, and three laser systems, including an argon (Ar)-ion laser and green and red helium–neon (HeNe) lasers or (2) a Leica SP5 CLSM equipped with a 63× glycerol-immersion objective (NA¼1.3) and five laser systems, including an Arion laser, green, orange, and red HeNe lasers, and a Radius 405-nm laser (Leica Microsystems). Green fluorescence was captured with an excitation wavelength of 488 nm and an emission band capturing wavelengths from 503 nm to 524 nm. Red fluorescence was captured with an excitation wavelength of 543 nm and an emission band capturing wavelengths from 600 nm to 625 nm. Chloroplast autofluorescence was captured with an emission band capturing wavelengths from 650 mm to 750 nm and subsequently pseudo-coloured blue. Images of leaf cells were acquired as single optical sections (i.e., z-sections) and saved as 1024 × 1024 pixel digital images. Excitations and emission signals for fluorescent proteins and chloroplasts were collected sequentially in double- or triple-labeling experiments and are the same as those described previously; single-labeling experiments showed no detectable crossover at the settings used for data collection. All of the fluorescence images of *N. benthamiana* plants were replicated across separate leaf samples with a minimum of three biological replicates. Each microscopy image included in the figures represents the typical patterns of fluorescence observed across the cells transiently expressing the selected exogenous gene(s) 48 h after infiltration. *N. benthamiana* leaf tissue under examination was taken at least 5 mm from the site of infiltration. Quantitative and qualitative data, such as fluorescence intensity and subcellular localization patterns, were collected and analyzed using the LAS AF and LAS X Life Science Microscope Software Platform version 1.4.6 28433 from Leica Microsystems.

## 3. Results

### 3.1. p21 Encoded by ORF8 Is a Cytosolic Protein

The predicted structure of p21 was analyzed by APBS and indicated that the majority of the putative protein surface is negatively charged (Figure 2A). p21 is fused to EGFP and transiently expressed in *N. benthamiana* localized to the cytosol in a diffuse pattern similar to free mRFP (Figure 2C,D).

### 3.2. p20A Encoded by ORF9 Is Likely Associated with Microtubules

The surface electrostatics of the predicted structure of p20A showed that several facets of the protein surface are positively charged (Figure 3A). p20A, when transiently expressed in *N. benthamiana*, co-localizes with host microtubule networks. That is, *N. benthamiana* co-infiltrated with p20A fused to EGFP (p20A-GFP) and C-terminal ER retention signal HDEL fused to monomeric Red Fluorescent Protein (mRFP) (mRFP-HDEL) yielded a green fluorescence pattern defined by punctate structures varying in size (Appendix A) [50]. To further investigate this apparent punctate pattern of aggregation that might be an artifact caused by the tendency of EGFP to form oligomers, p20A was fused to mRFP (p20A-mRFP). Initial observations revealed a fibrillar fluorescence pattern spanning the length of pavement cells. To identify cell organelles that p20A may associate with, p20B-mRFP was co-infiltrated with a GFP-tagged marker specific for mitochondria (CD3-987, targeting the inner membrane of the mitochondria), actin filaments (mTalin-GFP), or microtubules (MBD-GFP, the microtubule-binding domain of microtubule-associated protein 4 labeled with green fluorescent protein) (Figure 1B) [51,52,53]. In *N. benthamiana* leaves, p20A-mRFP exhibited a cytosolic fibrillar fluorescence pattern and did not colocalize with mitochondria or actin filaments (Figure 3C,D). Interestingly, when p20A-mRFP fluorescence was compared to MBD-GFP, an interchangeable pattern of fluorescence and clear examples of co-localization were observed in portions of the stochastic cytoskeletal network (Figure 3E) [54], suggesting that p20A likely associates with microtubules.

### 3.3. p20B Encoded by ORF10 Localizes to the Nucleus

The surface electrostatics of the predicted p20B structure show a group of five positively charged amino acids at the polar end of the protein (Figure 4B). In line with structural prediction, p20B-EGFP localized to the nucleus when transiently expressed in *N. benthamiana* and pavement cell nuclei were especially well highlighted by the fluorescence pattern due to the expression of p20B-EGFP (Figure 4C). To further support this, when co-infiltrated with a nuclear mRFP marker, a clear color overlay was observed, only in the nucleus, when the two channels were merged (Figure 4D). 

The p20B-EGFP fusion protein is approximately 49k Da, which is smaller than the 60 kDa size exclusion limit of the nuclear pore complex and thus, it is possible for the fusion protein to diffuse freely between the cytoplasm and the nucleus [55,56]. To verify that a protein is being trafficked into the nucleus, previous studies have fused a protein of interest to a bulky inert tag, such as beta-glucuronidase (GUS) weighing 69.4 kDa [57]. Following this methodology, a p20B-EGFP-GUS construct was generated to yield a recombinant protein of approximately 118 kDa (Figure 1B). The p20B-EGFP-GUS construct was then co-infiltrated with the coat protein of grapevine rupestris stem pitting-associated virus fused to mRFP (pCP:mRFP), a previously characterized nucleus-localized protein [58]. Captured with CLSM, large GFP fluorescent punctate bodies were observed in the pavement cells of co-infiltrated *N. benthamiana*. Many of these bodies were observed around or overlapping the pCP:mRFP-highlighted nucleus, indicating that p20B-EGFP-GUS was able to enter the nucleus (Figure 5A). 

The p20B amino acid sequence was submitted to Plant-mPLoc, which predicted p20B to localize to the nucleus, based on its current database of plant proteins [46]. The sequence was then submitted to the Identification Nucleus Signal Peptide (INSP) software, which uses existing protein structural knowledge and machine learning algorithms to identify localization signals within amino acid sequences [47]. INSP predicted a putative nuclear localization signal (NLS) peptide in p20B at amino acid positions 109–120 (EVCRGKRGSKKY), with a localization score of 0.93, indicating a very high likelihood that p20B localizes to the cell nucleus (Figure 4B). To test whether the predicted NLS sequence is indeed responsible for nuclear targeting, this annotated signal sequence was mutated using SDM in two steps, generating p20B-ΔNLS2-EGFP-GUS and p20B-ΔNLSε-EGFP-GUS constructs (Figure 1B). p20B-ΔNLS2-EGFP-GUS involved mutating the two lysine residues at positions 118 and 119 to alanine, which was then transiently expressed in *N. benthamiana* alongside pCP:mRFP. The fluorescence pattern of large punctate bodies was still observed, similar to p20B-EGFP-GUS (Figure 5B). Further mutations were generated in p20B-ΔNLSε-EGFP-GUS wherein positively charged amino acids R112, K114, and R115 of the putative NLS were converted to alanine. When transiently expressed in *N. benthamiana* alongside pCP:mRFP, the fluorescence pattern of p20B-ΔNLSε-EGFP-GUS was distinct from that of p20B-EGFP-GUS: there were no observable punctate bodies and the fluorescence pattern was diffuse throughout the transvacuolar and cortical cytosol as well as the perinuclear space of the cell, yet this was clearly excluded from the nucleus (Figure 5C).

To further validate this, the putative p20B NLS (MVCRGKRGSKKY) was amplified from the infectious clone of GLRaV-3 and tagged with EGFP and GUS to generate NLS-EGFP-GUS with a molecular weight of 96.81 kDa, which exceeds the nuclear exclusion limit (Figure 1B). This construct was transiently co-expressed in *N. benthamiana* alongside pCP:mRFP. Fluorescence was seen diffusely throughout the cytoplasm as well as the nucleus. A color shift was observed in the nucleus between the two fluorescent constructs (Figure 5D).

## 4. Discussion

### 4.1. Roles of Closteroviridae Genes in Unique Gene Blocks

The replicative and quintuple gene blocks of *Closteroviridae* have been well characterized in members of the *Closterovirus* genus [15]. Their gene functions and expression strategies are likely shared between *Closteroviridae* viruses, with minimal re-arrangements or deletions of ORFs. In contrast, the UGB, by its namesake, shows significantly more variation across *Closteroviridae* viruses. The number, size, sequence, and putative functions carried out by ORFs in UGB vary greatly. To date, ORFs in the UGB in the genomes of other *Closteroviridae* members function in suppressing the host RNA silencing system, systemic spread of the virus throughout the host, virus accumulation, and replication enhancers [23,24,25,59,60].

### 4.2. Open Reading Frame 8 Encodes p21, A Cytosolic Protein

There are numerous possibilities for the function of p21 inside the cytoplasm. Given the minimalistic biology of viruses, the abundance of sgRNA for ORF8 and the metabolic burden to conserve the gene sequence among GLRaV-3 isolates implies that the gene imbues the virus with a significant advantage in infectivity. In silico, it is revealed that p21 likely forms an oligomeric complex, although self-interaction has not been shown experimentally to date.

### 4.3. Microtubule Dynamics and Exploitation by Plant Viruses

Based on findings from this study, p20A is associated with cortical microtubules. The structure and function of microtubules is well conserved across all eukaryotic life forms. Several plant viruses have known associations with microtubules to facilitate cell-to-cell spread, anchoring viral replication complex (VRC) to the cortical region and their maturation, trafficking, and even inter-organism transmission [61]. For example, the movement protein (MP) of tobacco mosaic virus (TMV-MP) interacts with both the ER and microtubules [62] and is necessary for cell-to-cell viral movement [63]. The MP anchors maturing TMV VRCs to the ER membrane, after which they are released and transported to neighboring cells through plasmodesmata [64]. The MP of tomato mosaic virus Ob, TGB1 of potato mop-top virus, the CP of potato virus X, MP of grapevine fanleaf virus, and, with this study, now p20A of GLRaV-3 all associated with microtubules [65,66,67,68,69].

As we continue to deepen our understanding of plant cells, the study of microtubules has become an increasingly focused field in plant biology. For instance, a study of *Arabidopsis thaliana* discovered that over 100 different proteins bind and interact with microtubules [70]. Several microtubule-bound signaling proteins become active upon their release from depolymerizing microtubules. These signaling proteins have a wide array of cellular roles, such as the regulation of gene expression, translation, metabolism, cell wall modification, external environment manipulation, and environmental changes (e.g., temperature, osmolytes, heavy metal concentrations, and the presence of pathogens) [71,72,73,74,75,76,77,78,79]. To further complicate these interactions, the protein domains that mediate associations with microtubules are tremendously variable. The one commonality shared by microtubule-associating proteins is clusters of positively charged residues that are exposed on a protein surface, which facilitate interactions with the negatively charged termini of α- and β-tubulin dimers [80]. Proteins that interact with microtubules often have several groupings of positively charged exposed residues, as reported here for p20A [80] (Figure 3A,B).

The transient nature of microtubules is mirrored by the proteins that bind to them. These interactions are temporary to allow proteins and complexes to translocate along the length of microtubules or from one to another [81]. This can make live cell imaging of microtubule-associated proteins prone to artifacts, which is further compounded by interactions with fusion tags. EGFP is an exceedingly common fluorescent protein widely used in molecular biology labs. It folds efficiently and elicits robust and consistent fluorescence. However, EGFP is capable of low-affinity anti-parallel dimerization and is unsuitable for the study of proteins that associate with biological membranes or biopolymers such as the cytoskeleton network [82,83,84,85]. In a study by Snapp (2003), the expression of resident endoplasmic reticulum proteins tagged with EGFP was sufficient to induce alterations in the ER network, producing organized smooth ER structures called karmellae, whorls, and crystalloids. The fluorescence pattern of p20A-EGFP, in *N. benthamiana*, ranging from punctate structures of 1.25 μM down to sizes indistinguishable from diffuse fluorescence characteristic of the cytosol, was representative of aggregations (Appendix A). This was likely due to the EGFP tag disrupting the biologically relevant association of p20A with microtubules. Subsequent co-infiltrations of p20A-EGFP using significantly diluted infiltration cultures showed that a fluorescent fibrillar network pattern was clearly evident, further supporting this reasoning (Appendix A).

Plants utilize microtubules to generate and maintain their cell walls. Cellulose synthase complexes travel along cortical microtubules, depositing cellulose microfibrils through the plasma membrane to form networks interwoven into xyloglucan and pectin polysaccharide matrices [86]. Similar to this process of cell wall deposition, plants also use microtubules to deposit callose to the plasmodesmata under times of stress and during pathogen attack [87]. Callose is deposited by callose synthase, which is carried and inserted onto the plasma membrane by microtubules [86]. This process is especially vital to phloem tissues as callose can plug plasmodesmata that connect companion cells and the sieve elements following pathogen invasion or injury to prevent the spread of a pathogen [88,89]. Callose deposition is also a likely mechanism against GLRaV-3 infection, as expression of the callose synthase gene (beta-1,3-glucan or CalS1) is upregulated [90]. Notably, callose accumulation relies upon the proper function of cortical microtubules. Hence, p20A may enable the trafficking of GLRaV-3 virions or ribonucleoprotein complexes during systemic infection by overriding innate intracellular microtubule dynamics, which are essential to processes such as pathogen-responsive callose deposition.

### 4.4. The Viral RNA Silencing Suppressors of Closteroviridae

We have shown here that p20B, previously shown to serve as a VRSS [28], is a VRSS that localizes to the nucleus. Unique viral genes from members of the *Closterovirus* genus of the family *Closteroviridae* have been more thoroughly examined. For example, the citrus tristeza virus contains two distinct VRSS in its UGB, p23 and p20 [91]. p23 accumulates in three distinct areas when transiently expressed in *N. benthamiana*: nucleolus, Cajal bodies, and plasmodesmata, where it binds to ssRNA and dsRNA cooperatively, independent of sequences [92]. Beets yellows virus (BYV), another member of the *Closterovirus* genus, contains two genes in its unique gene block, p20 and p21, which are responsible for systemic movement and VRSS activity, respectively [93]. p20 accumulates in the plasmodesmata and binds to Hsp70h and the virion through an undetermined mechanism to translocate BYV from cell to cell and long distances across the phloem [94]. In contrast to p20 in BYV, p21 has a wealth of experimental data speaking to its mechanism of RNA silencing suppression [95]. Crystal structures of this protein show that it forms an octameric ring, with a central channel that is 90 Å wide. The inner surface of the oligomer carries a positive charge able to bind nucleic acids [93,95]. In addition to RNA silencing suppression, this cytosolic protein is required for the amplification of the genome of BYV and is abundant during early infection [96].

Nucleus-targeted VRSSs originating from *Closteroviridae* UGBs, as well as those from outside of *Closteroviridae,* require an NLS. NLSs refer to short stretches of amino acid residues that mediate the transport of proteins into the nucleus via importins [97]. Classical nuclear localization signals are categorized as monopartite or bipartite and are defined by an amino acid sequence of K-K or R-X-K/R or R/K-X10-12-K-R-X-K, respectively [98]. The putative NLS of p20B does not strictly fit into either of these categories and could thus be called a non-classical NLS (ncNLS). 

p20B, tagged with EGFP and GUS, is then transiently expressed in *N. benthamiana* and exhibits patterns of aggregations possibly indicative of disruption of p20B oligomerization, EGFP dimerization, or the increased strain placed on protein folding pathways by over-expression of these large composite proteins, leading to misfolding. When the p20B NLS signal was tagged to a fusion of EGFP and GUS, a composite protein with a molecular weight far beyond the SEL of nuclear pore complexes with clear co-localization with a nuclear marker was seen.

The in silico predicted structure and subsequent analysis of p20B suggest the possibility that it may participate in RNA silencing as reported previously [28]. A predicted hexameric state of the protein has a central channel 20 Å in diameter, which is equivalent to the average size of dsDNA (i.e., 20 Å [99]; Appendix A). The putative NLS sequences are organized onto one side of the ring structure (Appendix A). This suggests that the complex may interact with or sequester nucleic acid chains involved in host RNA silencing. Alternatively, p20B may bind and sequester components of the RNA silencing pathway in the nucleus such as AGO proteins, Hsp90, or cochaperones necessary for RISC maturation. Follow-up recombinant protein–protein binding assays are necessary to provide a clear picture of cellular interaction partners. GLRaV-3 is a serious concern for the global grape and wine industries. Dated screening and cumbersome management methods are the primary strategies to combat the virus. The findings presented here shed light on the relationship between the host and virus, building our foundational understanding and informing the development of strategies to manage and remedy GLRaV-3 infection.

## 5. Conclusions

This study advances our understanding of the UGB of GLRaV-3 by elucidating the localization and emphasizing the distinct roles of specific proteins involved in viral infectivity and host interaction. Here, we identify p21 and p20A as cytosolic and cortical microtubule localized proteins, respectively, and propose mechanisms in which their organization may facilitate viral access to the phloem for systemic movement. Furthermore, we report that p20B contains a non-classical NLS and exhibits nuclear trafficking, indicative of the involvement of nuclear entry in its function in suppressing host RNA silencing machinery. Future research includes elucidating interactors of p21, p20A, and p20B in host systems to uncover their roles in viral pathogenicity and virus-host interaction pathways. These findings provide critical insights into the molecular mechanisms employed by GLRaV-3 and underscore the diverse functionalities encoded within the UGBs of the *Closteroviridae* family.

## Figures and Tables

**Figure 1 biomolecules-14-00977-f001:**
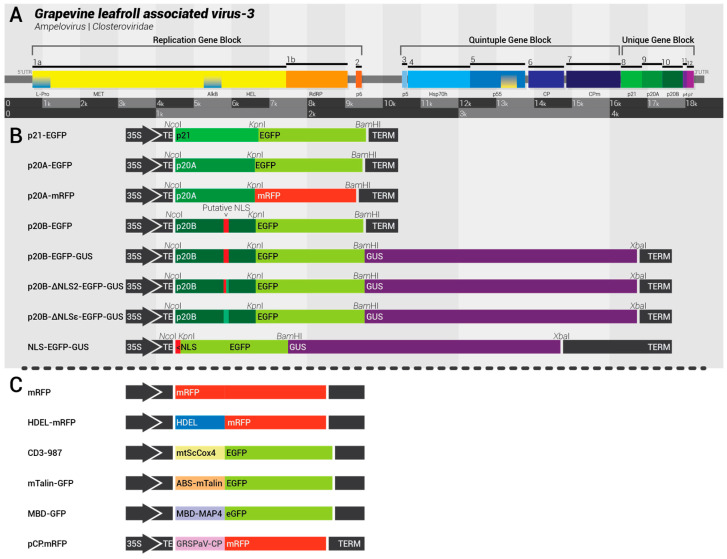
Genome structure of grapevine leafroll associated virus 3 (GLRaV-3) and protein expression constructs. (**A**). Genome structure of GLRaV-3. The RNA genome contains 12–13 open reading frames (ORFs). ORF1a/b encodes a replicase polyprotein. ORFs 3–7 make up the “Quintuple Gene Block” (QGB) shared among the family *Closteroviridae*, which encode five polypeptides designated p5, Hsp70h, p55, the coat protein, and the coat protein minor. ORFs 8–12 make up the “Unique Gene Block” (UGB), which encodes the polypeptides p21, p20A, p20B, p4, and p7. (**B**). Graphic representation of the viral protein expression constructs. All constructs were made based on pRTL3-GFP and pRTL3-mRFP. Each contains the cauliflower mosaic virus (CaMV) 35S promoter, the tobacco etch virus translation enhancer (TE), and the CaMV 35S termination signal (TERM). Restriction sites used to make these constructs are given above each construct. GFP: green fluorescent protein; mRFP: monomeric red fluorescent protein. (**C**). Graphic representation of the organelle and cytoskeletal fluorescent markers used in this study.

**Figure 2 biomolecules-14-00977-f002:**
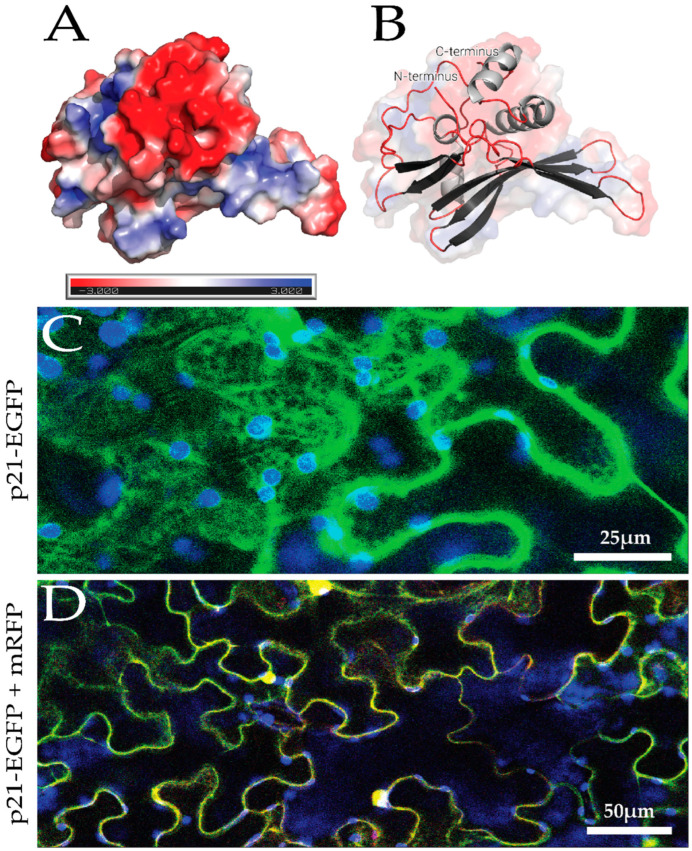
In silico analysis of the predicted structure and subcellular localization of p21 using CLSM. In silico protein structural prediction and analysis of p21 as predicted by ColabFold. (**A**). Surface electrostatics of p21 generated by APBS (Jurrus E et al.). (**B**). Cartoon representation of predicted p21 structure. N and C termini are labeled. (**C**). *N. benthamiana* agro-infiltrated with p21-EGFP. (**D**). *N. benthamiana* agro-infiltrated with p21-EGFP and mRFP.

**Figure 3 biomolecules-14-00977-f003:**
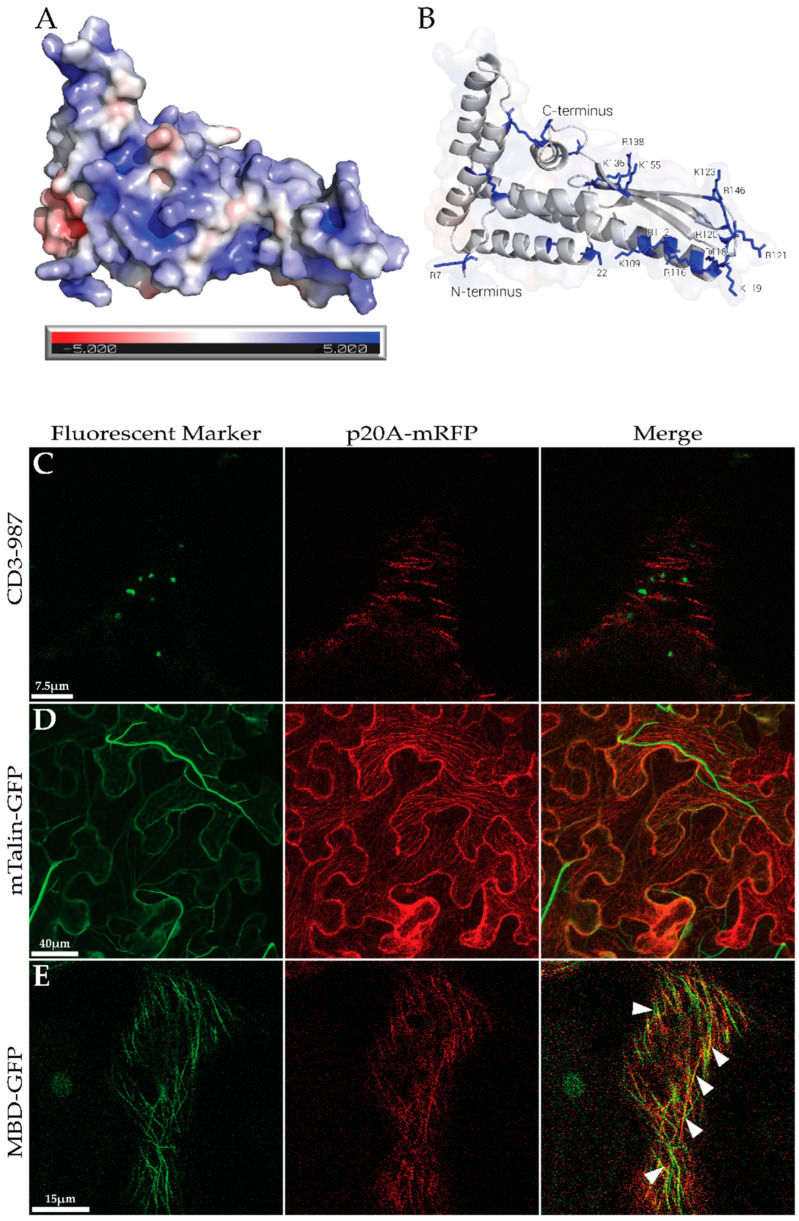
In silico analysis of the predicted structure and subcellular localization of p20A using CLSM. (**A**). Predicted structure of p20A, colored by electrostatics generated by APBS. (**B**). Cartoon representation of the predicted structure of p20A. Positively charged residues are colored blue. N and C termini are labeled. (**C**). *N. benthamiana* leaf tissue agro-infiltrated with CD3-987 and p20A-mRFP. Green and red channels are overlayed in the rightmost panel. (**D**). *N. benthamiana* leaf tissue agro-infiltrated with mTalin-GFP and p20A-mRFP. Green and red channels are overlayed in the rightmost panel. (**E**). *N. benthamiana* leaf tissue agro-infiltrated with MBD-GFP and p20A-mRFP. Green and red channels are overlayed in the rightmost panel. White triangles indicate examples of colour shift in overlayed image.

**Figure 4 biomolecules-14-00977-f004:**
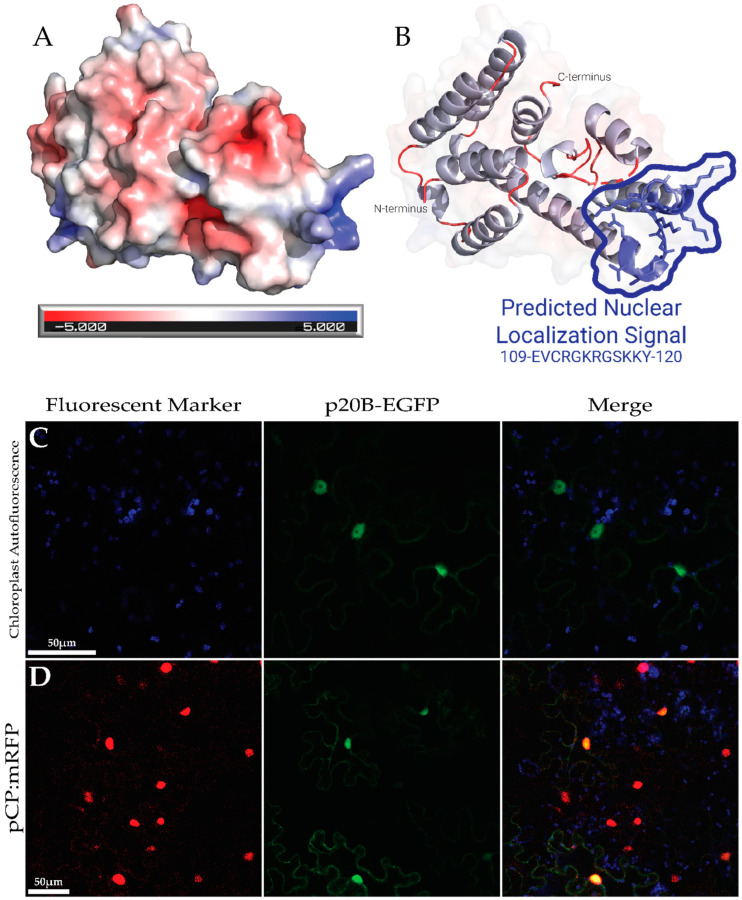
In silico analysis of the predicted structure and subcellular localization of p20B using CLSM. (**A**). Predicted structure of p20B, colored by electrostatics calculated by APBS. (**B**). Cartoon representation of the predicted structure of p20B. Predicted nuclear localization signal is colored in blue and highlighted. N and C termini are labeled. (**C**). Chloroplast autofluorescence captured from *N. benthamiana*. agro-infiltrated with p20B-EGFP. Chloroplast autofluorescence and green channels are overlayed in the rightmost panel. (**D**). *N.benthamiana* leaf tissue infiltrated with pCP:mRFP and p20B-EGFP. Chloroplast autofluorescence; red and green channels are overlayed in the rightmost panel.

**Figure 5 biomolecules-14-00977-f005:**
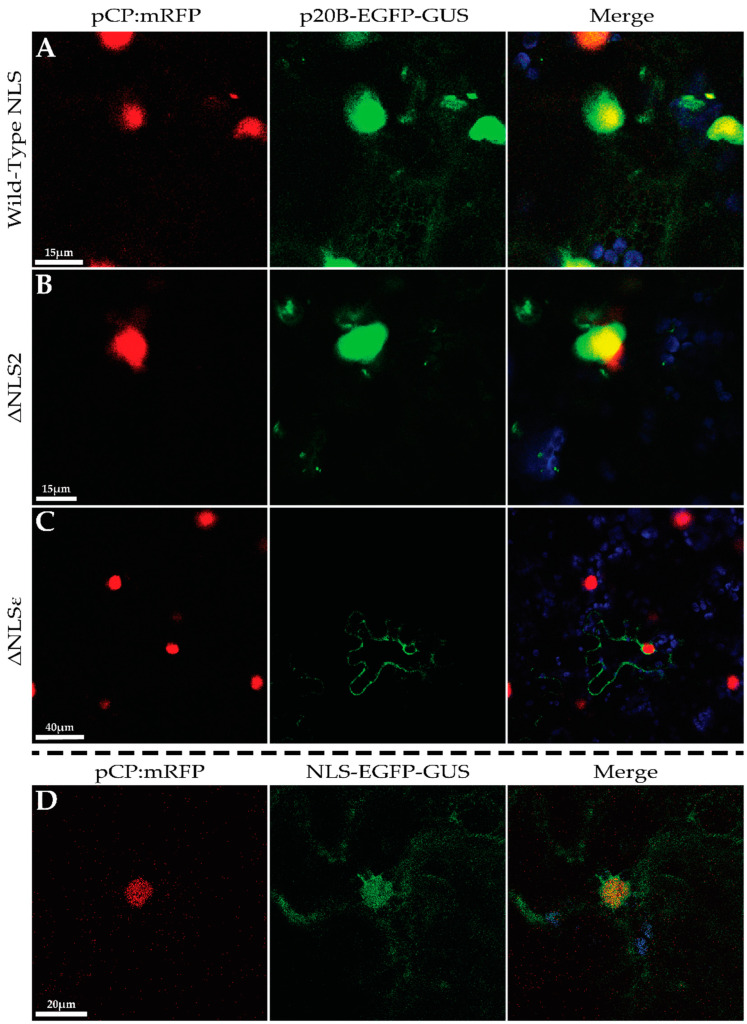
p20B GUS tagging, p20B NLS mutagenesis, and p20B NLS tagging of subcellular localization analysis using CLSM. (**A**). *N. benthamiana* agro-infiltrated with pCP:mRFP and p20B-EGFP-GUS. Chloroplast autofluorescence; red and green channels are overlayed in the rightmost panel. (**B**). *N. benthamiana* agro-infiltrated with pCP:mRFP and p20B-ΔNLS2-EGFP-GUS. Chloroplast autofluorescence, red and green channels are overlayed in the rightmost panel. (**C**). *N. benthamiana* agro-infiltrated with pCP:mRFPand p20B-ΔNLSε-EGFP-GUS. Chloroplast autofluorescence; red and green channels are overlayed in the rightmost panel. (**D**). *N. benthamiana* agro-infiltrated with pCP:mRFP and NLS-EGFP-GUS. Chloroplast autofluorescence, red and green channels are overlayed in the rightmost panel.

**Table 1 biomolecules-14-00977-t001:** Primers used in making gene expression constructs, site-directed mutagenesis, and the amplification of the putative NLS of p20B.

Primers	Sequence	Purpose
ORF8amp-F	5′-CCATGGAATTCAGACCAGTTTTAATTAC-3′	Amplify ORF8. Add an NcoI site to the 5′ end and KpnI to the 3′ end.
ORF8amp-R	5′-GGTACCTTTAAACCCTGGGTAATCAG-3′	
ORF9amp-F	5′-CCATGGCCAGGTTACTTTCGCTCCGC-3′	Amplify ORF9. Add an NcoI site to the 5′ end and KpnI to the 3′ end.
ORF9amp-R	5′-GGTACCACCTAGAGATGAGATTAGTA-3′	
ORF10amp-F	5′-CCATGGACCTATCGTTTATTATCGTG-3′	Amplify ORF10. Add an NcoI site to the 5′ end and KpnI to the 3′ end.
ORF10amp-R	5′-GGTACCCAGCGCTCCGCAACAAAGCGT-3′	
GUSamp-F	5′-AAAAGGATCCTTACGT CCT GTA GAA ACCCCAACCC-3′	Amplify GUS. Add an BamHI site to the 5′ end and Xbai to the 3′ end.
GUSamp-R	5′-AAAATCTAGAAAATCATTGTTTGCCTCCCTGCTGC-3′	
ΔNLS2-F	5′-GGGAAGTGCAGCATATCTTGGATACTTAAGTGA TAAATGCTCTGGCAAACATATAATGCTAACTCAG-3′	Site directed mutagenesis primers were used to mutate K118A and K119A in the putative p20B NLS.
ΔNLS2-R	5′-CCAAGATATGCTGCACTTCCCCTCTTTC CACGACACACTTCGAATAACTCCAT-3′	
ΔNLSε-F	5′-TGTGCAGGAGCAGCAGGAAGTGCAGCATATCTT GGATACTTAAGTGATAAATGCTCTGGCAAACATA-3′	Site directed mutagenesis primers were used to mutate R112A, K114A and R115A in the putative p20B NLS.
ΔNLSε-R	5′-TTCCTGCTGCTCCTGCACACACTTCGAATAAC TCCATCGACTTGATGACTTCCAGAACGTCTTC-3′	
NLSamp-F	5′-AAACCATGGTGTGTCGTGGAAAGAG-3′	Amplify the putative NLS of ORF10. Add an NcoI site and methionine to the 5′ end and KpnI to the 3′ end.
NLSamp-R	5′-AAAGGTACCATATTTTTTACTTCCCCTCTTTCC-3′	

## Data Availability

Gene constructs that were made pursuant to this study and are available upon request.

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
