# Peer review of "Elucidating the Subcellular Localization of GLRaV-3 Proteins Encoded by the Unique Gene Block in N. benthamiana Suggests Implications on Plant Host Suppression"

_biomolecules, 2024, doi:10.3390/biom14080977_

Round 1

Reviewer 1 Report

Comments and Suggestions for Authors

I recommend that the authors briefly describe the modes of GLRaV-3 transmission, including insect vectors in the introduction.

Fig4

The authors assert that p20B protein is localized to the nucleus by detecting chloroplast fluorescence. I cannot understand why can increased fluorescence in the chloroplasts prove localization of the protein to nucleus? Please explain this clearly.

Line 453

The authors insist that p20B protein of GLRaV-3 is an RNA silencing suppressor based on two factors: in silico, it retains high homology with RNA silencing suppressors of other viruses, and the p20B protein localizes to the nucleus of N. benthamiana cells. However, to asset that this protein is definitely silencing suppressor, it is necessary to verify whether the silencing would be virtually compromised when GLRaV-3 p20B is introduced into the plants expressing transgene induced RNA silencing. At this point, the authors can only say there is a possibility that the protein may be RNA silencing suppressor.

That is all.

Author Response

Comments 1: I recommend that the authors briefly describe the modes of GLRaV-3 transmission, including insect vectors in the introduction.

Response 1: Thank you for pointing this out. I agree with this comment. Therefore, I have added a brief explanation of modes of GLRaV-3 transmission and mentioned Pseudococcidae mealybugs are the main insect vector. This addition can be found on page 1, Paragraph 2 and line 42.

” The virus is transmitted through infected propagation materials and Pseudococcidae mealybugs [3,4].” 

Comments 2: Fig4. The authors assert that p20B protein is localized to the nucleus by detecting chloroplast fluorescence. I cannot understand why can increased fluorescence in the chloroplasts prove localization of the protein to nucleus? Please explain this clearly.

Response 2: Thank you for pointing this out. This figure may be misleading. Chloroplast autofluorescence was shown as a reference point for plant cell contours and physiological condition. Figure 4C was taken to show the fluorescence pattern of p20B-EGFP when infiltrated individually. Figure 4D further verifies the nuclear localization of p20B-EGFP by co-infiltrating it with the established nuclear localizing protein pCP:mRFP. In the merged panel of 4D chloroplast autofluorescence is shown again for the same purpose. We would like to point out that using chloroplast autofluorescence as a reference point is commonly used by other researchers in cell biology.

Comments 3: Line 453. The authors insist that p20B protein of GLRaV-3 is an RNA silencing suppressor based on two factors: in silico, it retains high homology with RNA silencing suppressors of other viruses, and the p20B protein localizes to the nucleus of N. benthamiana cells. However, to asset that this protein is definitely silencing suppressor, it is necessary to verify whether the silencing would be virtually compromised when GLRaV-3 p20B is introduced into the plants expressing transgene induced RNA silencing. At this point, the authors can only say there is a possibility that the protein may be RNA silencing suppressor.

Response 3: The findings presented in this paper do not demonstrate that p20B is an RNA silencing suppressor. That p20B served as a RNA silencing suppressor was demonstrated in a 2012 study by Gouveia et al. This has been accepted as common knowledge in the published literature. We referenced this study in the introduction on line 102. “p20B has also been identified as a viral RNA silencing suppressor (VRSS) in N. benthamiana and a potential determinant of pathogenicity [26].” The study showed p20B was a suppressor through the method that you have mentioned. Additionally, we did not find that the in-silico prediction of p20B retains homology with other established RNA silencing suppressors.

Reviewer 2 Report

Comments and Suggestions for Authors

The manuscript titled as Elucidating the Subcellular Localization of GLRaV-3 Proteins 2 encoded by the Unique Gene Block Suggests Implications on 3 Plant Host Suppression, focused on the role of GLRaV-3 ORF8, ORF9 and ORF10 which encode the proteins p21, p20A and 18 p20B, respectively. The authors concluded that p21 localizes to the cytosol,p20A associates with microtubules and p20B is trafficked into the nucleus to carry out the suppression of host RNA silencing.

General Comment: The authors did very good job in performing  the experiments and putting them on paper.

There is need to make few minor clarifications and amendments before further processing of the manuscript.

The title of the study need to be adjusted as it give the feel that grapevine was mainly used to conduct the study instead all the assays were performed in N. benthamiana.

Although introduction is very detailed but for a manuscript it should be concise. I will suggest the authors to considerably reduce the length of the introduction. There is no need to use the subheadings under the introduction section.

There is need to comprehensively review that part. In addition a preoper novelty statement is required.

Methodology section is well written and properly referenced.

Please also explain/justify the use of N. benthamiana. 

What are the possibilities of performing the same tasks using actual host of the virus.

In results, some of the figures pixel quality need to be checked.

Results are very well written, while discussion can be amended with economic significance of the work.

Discuss role and potential use of your findings in understanding host, virus complex and importance/utility of the information in designing future plant protection strategies.

 Reference formatting should be uniform. Some of the references cited have no DOI number mentioned with them. 

Best Wishes

Author Response

Comments 1: The title of the study need to be adjusted as it give the feel that grapevine was mainly used to conduct the study instead all the assays were performed in N. benthamiana.

Response 1: I agree with this comment. Therefore we have changed the title in the revised manuscript as follows
Elucidating the Subcellular Localization of GLRaV-3 Unique Gene Block in N. benthamiana Suggests Implications on Plant Host Suppression.

Comments 2: Although introduction is very detailed but for a manuscript it should be concise. I will suggest the authors to considerably reduce the length of the introduction. There is no need to use the subheadings under the introduction section. There is need to comprehensively review that part.

Response 2: Thank you for pointing this out. I agree and have removed the subheadings in the introduction and further condensed the introduction.

In additional to sentence restructuring, I have removed the following from the introduction:
Tartaric acid, malic acid, and total organic acids are reduced [9]. Furthermore, valine, methionine, and glutamic acid levels are altered in infected grapes.
Stalled fermentation produces unwanted compounds like sulfur, fusel alcohols, and thiol-containing compounds in wine
RNA silencing plays a multifaceted role in plant antiviral defense.
Representing a major form of innate immunity in plants and invertebrates.
By infiltrating each p21, p20A, and p20B tagged with Enhanced Green Fluorescent Protein (EGFP), we may narrow down the potential described mechanisms or gain in-sight into a novel mechanism through which these viral proteins suppress and manipulate the host cell.

Comments 3: In addition a proper novelty statement is required.
Response 3: Thank you for pointing this out. I agree with this comment. I have restructured the final paragraph of the introduction in order to reinforce the novelty of the report and describe its impact on the field.

Comments 4: Please also explain/justify the use of N. benthamiana. 
Response 4: Thank you for pointing this out. We agree with your comment and have added justification to the methods section
This change can be found on page number 4, paragraph 4 and line 180.
“N. benthamiana was chosen as the model system for our study as it is amenable to agrobacterium mediated infiltration, susceptibility to a wide range of viruses, and grows rapidly in controlled environments offering quick turn around time for experimentation [47].” This is a model experimental system that are commonly used to study plant viruses.

Comments 5: What are the possibilities of performing the same tasks using actual host of the virus.
Response 5: I would be very interested in repeating these experiments in Vitis spp., but due to the extended time it takes for perennial woody grapevines to reach maturity and the lack of a robust method for transient expression this would be very difficult to conduct.

It is worth noting that the tubulin and components of the RNA silencing pathway are well conserved amongst higher plants, and thus N. benthamiana serves as a robust experimental organism for our purposes.

Comments 6: In results, some of the figures pixel quality need to be checked.
Response 6: Can you indicate which figures you are seeing quality issues with? After looking at the pngs I used for each figure, they are all well above 300dpi, and should not have any issues with low resolution.

Comments 7:
Results are very well written, while discussion can be amended with economic significance of the work. Discuss role and potential use of your findings in understanding host, virus complex and importance/utility of the information in designing future plant protection strategies.
Response 7: Thank you for your feed back. We have concluded the discussion by reiterating the economic significance of GLRaV-3 on the grape and wine industry. The findings in this report are largely foundational and I have emphasized their importance in the development of management strategies.

Page 16, Paragraph 1, Line 502
"GLRaV-3 is a serious concern for the global grape and wine industries. Dated screening and cumbersome management methods are the primary strategies to combat the virus. The findings presented here reveal shed light on the relationship between host and virus, building our foundational understanding and informing the development of strategies to manage and remedy GLRaV-3 infection."

Comments 9: Reference formatting should be uniform. Some of the references cited have no DOI number mentioned with them. 
Response 9: Thank you for pointing this out. I have gone through my references and updated the missing information and doi numbers in the reference management software.

Round 2

Reviewer 1 Report

Comments and Suggestions for Authors

The matters I pointed out previously have been appropriately corrected. Therefore, I recommend that this paper would be published in the Biomolecules journal.